# A Comparison of Spinopelvic Alignment and Quality of Life between Farmers and Non-Farmers: A Cross-Sectional Population-Based Study in a Japanese Rural Area

**DOI:** 10.3390/jcm12041393

**Published:** 2023-02-09

**Authors:** Takahisa Ogawa, Kazuyuki Fukushima, Miyuki Niimi, Haggai Schermann, Takayuki Motoyoshi, Janelle Moross, Motonori Hashimoto, Takashi Hirai, Takeo Fujiwara, Atsushi Okawa, Yoshiro Kurosa, Toshitaka Yoshii

**Affiliations:** 1Department of Orthopedic Surgery, Tokyo Medical and Dental University Graduate School of Medicine, Tokyo 113-8510, Japan; 2Department of Orthopedic Surgery, Saku General Hospital Saku Medical Center, Nagano 385-0051, Japan; 3Department of Health Promotion, Saku General Hospital Saku Medical Center, Nagano 385-0051, Japan; 4Division of Orthopedic Surgery, Tel Aviv Sourasky Medical Center, Tel Aviv University, Tel Aviv 6423906, Israel; 5Office for Global Education and Career Development, Institute of Global Affairs, Tokyo Medical and Dental University, Tokyo 113-8510, Japan; 6Department of Global Health Promotion, Graduate School of Medical and Dental Sciences, Tokyo Medical and Dental University, Tokyo 113-8510, Japan; 7Department of Orthopaedic and Spine Surgery, Graduate School of Medical and Dental Sciences, Tokyo Medical and Dental University, 1-5-45 Yushima, Bunkyo-ku, Tokyo 113-8519, Japan

**Keywords:** agriculture, farmer, kyphoscoliosis, quality of life

## Abstract

Background: It may be difficult to define what would constitute an abnormal spinal sagittal alignment. The same degree of malalignment may be found both in patients with pain and disability and in asymptomatic individuals. This study focuses on elderly farmers who characteristically have a kyphotic spine, in addition to local residents. It questions whether these patients experience cervical and lower back symptoms, respectively, more often than elderly people who never worked on a farm and do not have a kyphotic deformity. Previous research could have been biased by sampling patients who came to a spine clinic for treatment, whereas this study sampled asymptomatic elderly who may or may not have had kyphosis. Methods: We studied 100 local residents at their annual health checkup (22 farmers and 78 non-farmers) with a median age of 71 years (range 65–84 years). Spinal radiographs were used to measure sagittal vertical axis, lumbar lordosis, thoracic kyphosis and other measurements of sagittal malalignment. Back symptoms were measured using Oswestry Disability Index (ODI) and Neck Disability Index (NDI). The association between alignment measures and back symptoms were calculated by bivariate comparison between patient groups and by Pearson’s correlation. Results: About 55% of farmers and 35% of non-farmers had abnormal radiographs (i.e., vertebral fracture). Farmers had higher measurements of sagittal vertical axis (SVA), compared to non-farmers, when measured from C7 (median 24.4 mm vs. 9.15 mm, *p* = 0.04) and from C2 (47.65 vs. 25.3, *p* = 0.03). Lumbar lordosis (LL) and thoracic kyphosis (TK) were significantly decreased in farmers vs. non-farmers (37.5 vs. 43.5, *p* = 0.04 and 32.5 vs. 39, *p* = 0.02, respectively). The ODI was likely to be higher among farmers compared to non-farmers while NDI scores showed no significant difference between farmers and non-farmers (median 11.7 vs. 6.0, *p* = 0.06 and median 13 vs. 12, *p* = 0.82, respectively). In terms of correlation among spinal parameters, LL had a higher correlation with SVA, but TK had less correlation with SVA among farmers compared to non-farmers. There was no significant correlation between disability scores and measurements of sagittal alignment. Conclusions: Farmers had higher measurements of sagittal malalignment, characterized by loss of LL, decreased TK and an increased forward translation of cervical vertebrae relative to sacrum. ODI was likely to higher in farmers compared to non-farmers although the association did not reach a significant level. These results probably indicate that the gradual development of spinal malalignment in agricultural workers does not result in excess morbidity compared to controls.

## 1. Introduction

Spinal deformity refers to abnormal curvature of the spine and is readily noticeable as anterior tilting of the upper body with abnormal posture. Spinal deformity often progresses with age and predominantly affects elderly individuals. Since the 2010s, surgeons specializing in spinal surgery have made progress in the study of the association between sagittal alignment and symptoms related to the back [1]. A variety of factors are known to be associated with the development of kyphoscoliosis, including lumbar compression fractures and disc degeneration [2]. In addition, excessive load on spinal vertebrae due to heavy lifting can contribute to spinal malalignment [3,4,5].

Agricultural farming is a physically demanding job. Postures typically required during agricultural work apply significant loads to the back [6]. Therefore, we hypothesized that farmers have a greater degree of spinal malalignment and disability related to neck and back symptoms. As a wide range of sagittal alignment measurements have been reported in asymptomatic populations [7], the presence of kyphoscoliosis does not necessarily lead to severe lumbo-sacral pain and patients do not always present for medical care [8,9,10]. Thus, it is important to include the general population regardless of back symptoms in studies of spinal alignment. However, most previous cohort studies comprised patients with kyphoscoliosis presenting with complaints of back pain [5]. A previous study including only patients with spinal malalignment compared the degree of malalignment between farmers and non-farmers [11]. There have been no previous studies evaluating the association between agricultural work and spinal alignment in a general population including healthy participants.

Accordingly, the purpose of the present study was to investigate the effects of farming on sagittal alignment in a general population of individuals that included farmers presenting for a health checkup. We further evaluated the relationship between malalignment and symptoms related to the back and neck using patient-reported outcomes.

## 2. Materials and Methods

### 2.1. Study Design and Population

This was a cross-sectional study of 100 local residents aged 65 years or older who underwent a two-day health checkup between October 2020 and January 2021. Ethical approval of the present study was obtained from our institutional review board. All participants provided written consent prior to enrollment. We excluded patients who were unable to undergo standing full-spine radiographic examinations due to advanced dementia or lower limb paralysis.

### 2.2. Radiological Evaluations

On the first day of the health checkup, participants underwent full-spine radiography and a questionnaire survey. Anterior–posterior and lateral full-spine radiographs were taken in a simple, relaxed standing position. These were then reviewed for quality by a board-certified spinal surgeon and a full-time radiologist. Images of inadequate quality were immediately re-taken. Radiological parameters related to sagittal alignment were measured on full-spine lateral radiographs based on the Scoliosis Research Society (SRS)-Schwab radiological classification, including sagittal vertical axis (SVA), lumbar lordosis (LL), pelvic incidence (PI), PI-LL, Thoracic Kyphosis (TK), C2-7SVA, cervical lordosis (CL) and T1 slope (Figure 1). Parameters provided in Figure 1 were measured by a board-certified orthopedic surgeon. In addition, a board-certified orthopedic surgeon reported the presence of pathologic findings including morphological vertebral compression fractures and disc degeneration.

### 2.3. Clinical Evaluations

The questionnaire survey included the Neck Disability Index (NDI) and Oswestry Disability Index (ODI) [12,13,14,15]. In addition, questions regarding family structure, occupation and employment status were included in the medical questionnaire. Questionnaires were completed by all participants independently. A clinical research coordinator assisted some participants who had poor vision due to presbyopia or cataracts.

### 2.4. Statistical Analyses

Participant demographics, radiographic measurements and questionnaire scores were compared between farmers and non-farmers using the Wilcoxon rank-sum test and Fisher’s exact test based on the distribution of values. Correlations between radiographic parameters and ODI or NDI scores were calculated using the Spearman’s rank correlation coefficient. Data management was performed independently at an external data center. Statistical analyses were performed using JMP 5 software (SAS Institute, Cary, NC, USA) and Stata version 16.1 (Stata Corp, College Station, TX, USA). The probability of type I error was set to 0.05 for all statistical analyses.

## 3. Results

The background information of the study participants is shown in Table 1. The present study comprised a total of 100 participants including 60 males and 40 females, with a median age of 71 years (range 65–84 years). Farmers were significantly younger than non-farmers by an average of 3 years (*p* = 0.02, Wilcoxon rank-sum test). All farmers were still working while 62% of non-farmers were unemployed (*p* < 0.01, Fisher’s exact test; Table 1).

Abnormal radiographic findings were observed in 39 (39%) patients, with 13 (33%) vertebral compression fractures (Table 2). Abnormal radiographic findings were more common in farmers than in non-farmers (55% vs. 35%; *p* = 0.09).

Evaluations of sagittal alignment demonstrated that LL was significantly decreased in farmers compared with non-farmers (median, 37.5 mm vs. 43.5 mm; *p* = 0.04). SVA (median, 24.4 mm vs. 9.15 mm; *p* = 0.02) and C2-7SVA + SVA (median, 47.65 mm vs. 25.3 mm; *p* = 0.03) were significantly increased in farmers compared to non-farmers. Notably, thoracic kyphosis was significantly decreased in agricultural workers (Table 2). There was a trend toward higher ODI values in farmers compared to non-farmers (median, 11.7 vs. 6.0; *p* = 0.06), while no significant difference in NDI was observed between farmers and non-farmers (median, 13 vs. 12; *p* = 0.82; Table 2).

Table 3 and Table 4 show the radiographic parameters of all participants. SVA, LL and PI values were significantly correlated with PI-LL values. In addition, PI-LL, TK and C2–7 SVA values were significantly correlated with C2–7SVA + SVA values. Both T1 slope and cervical LL were significantly associated with all parameters other than PI-LL.

Table 5 and Table 6 show associations between radiographic parameters and ODI or NDI values in farmers and non-farmers. In farmers, only T1 slope demonstrated a significant negative correlation with ODI (correlation coefficient, −0.51), with no significant association observed between ODI and NDI values (correlation coefficient, 0.16). Conversely, in the non-farmer group, no significant correlation was observed between T1 slope and ODI (correlation efficient, −0.08). NDI values were significantly associated with ODI values (correlation coefficient, 0.57).

## 4. Discussion

The purpose of the present study was to evaluate the relationship between agricultural farming work and the prevalence and morbidity of spinal malalignment. In addition, we conducted a population-based study including local residents who were not known to have spinal kyphosis other than participants with a characteristic kyphotic deformity. The results of the present study demonstrate that farmers are more likely to have significant sagittal malalignment. Spinal deformity in farmers is characterized by anterior tilt of the upper trunk, apparently due to reduced LL rather than increased thoracic kyphosis. This finding is likely attributable to many years of agricultural work where the center of gravity is displaced forward due to frequent bowed posture and shoulder loading with compensation by the upper trunk. This type of spinal deformity differs from the sagittal malalignment observed in elderly osteoporotic patients who may have increased thoracic kyphosis (“hump”) due to vertebral fractures.

In the region where our study was conducted, the primary crops produced by farmers were vegetables and rice. Some farmers partially used machinery, but manual labor also made up a large portion of the work process. In vegetable and rice farming, work is mainly performed by bending the upper body toward the ground, not by lifting the upper limbs against gravity. While planting and harvesting rice, numerous farmers adopt a posture of maintaining an extended knee position while bending only at the waist, as a result of the burden on the knee joint. These postures are likely to strain the lower back and possibly cause kyphosis deformity of the lumbar spine [16].

Previous studies have evaluated farmers with back pain. In 2018, Jain et al. reported a prevalence of lumbar pain of 71.4% in 138 manual agricultural workers, which was more common than symptoms affecting other parts of the body [17]. In 2019, Khan et al. conducted a review on the relationship between posture during agricultural work and the development of lower back pain. Eight of the nine articles concluded that there was an association between posture and the onset of lower back pain. Notably, these studies used questionnaires but did not perform radiographic measurements [18]. The results of the present study corroborate the findings of previous studies demonstrating that kyphotic deformities may develop without underlying osteoporosis or compression fractures [2,19]. In the present study, 61 participants had normal spinal radiographs while the remaining 39 participants had abnormalities on spinal radiographs, with the majority (two-thirds) having disc degeneration rather than vertebral fractures.

Vertebral fracture is a known cause of spinal deformity and reportedly present in 36–38% of patients with severe kyphoscoliosis [2,20]. Insufficient back muscle strength may also affect sagittal malalignment. There is a significant correlation between erector spinae muscle density and severe kyphoscoliosis exceeding 40 degrees [21]. Degenerative disc disease also contributes to the progression of sagittal spinal malalignment. Disc desiccation results in height loss with increasing age, which is typically asymmetrical with significant anterior loss. As a result, anterior wedging may occur, leading to kyphosis progression [2,22]. Mann et al. conducted a retrospective study of 100 asymptomatic healthy women ranging in age from 39 to 91 years and reported a significant correlation between the angle of kyphosis and anterior disc height (r = 2.34) [22]. In addition, a population-based study in Southern California comprising 1407 community-dwelling ambulatory adults posited that hyperkyphosis in elderly patients is caused by degenerative disc disease, even among individuals without osteoporosis or vertebral fractures [2].

Of the above processes, disc degeneration is the most likely to be present among agricultural workers, with weakness of the erectus spinae muscles less likely to be present. The radiographic features observed in agricultural workers included significantly reduced LL and significantly decreased thoracic kyphosis compared to non-agricultural workers. Furthermore, SVA and C2-7SVA + SVA values were significantly higher in agricultural workers compared to non-agricultural workers. Presumably, agricultural workers lean forward while working and lifting heavy loads. As a compensatory mechanism for the postural abnormality, TK was significantly decreased in agricultural workers suggesting that decreases in LL precede decreases in TK. Furthermore, TK may decrease as an adaptation to decreases in LL. Indeed, the findings of the present study demonstrated a lower correlation between TK and SVA but higher correlation between LL and SVA in farmers compared to non-farmers. This adaptation may not necessarily be painful as there was no significant difference in NDI between farmers and non-farmers. A separate cross-sectional study of asymptomatic elderly adults reported an average SVA of 25 ± 32 mm, a similar value to the average SVA observed in farmers in the present study (Table 2).

Regarding the radiographic findings observed in farmers, we posit LL may decrease at a younger age due to overload from agricultural work followed by decreases in TK as the trunk tends to tilt forward to compensate for kyphosis. In other words, the kyphotic deformity observed in agricultural workers represents a special deformation due to long-term exposure to physical labor and may have a unique pattern that differs from kyphotic deformities observed in the general population. Zhang et al. conducted a study in a group of Chinese farmers and observed even more severe kyphotic deformities (SVA, 41.2 ± 59.0 mm) [5]. They also reported increased thoracic kyphosis with lumbar kyphosis in farmers compared to controls. This finding contradicts the decreased LL but decreased TK, likely as a compensatory mechanism, observed in the present study. This paradoxical result may be due to differences in the types of agriculture work undertaken in China and Japan, such as the use of machinery for rice agriculture versus the greater shoulder loading involved in corn agriculture. In order to mitigate the potential for stress on the lumbar region, it is crucial to adopt a posture in which the knee joints are flexed. Adopting such a stance may prevent the exacerbation of lumbar kyphosis resulting from degeneration of the intervertebral discs [23]. In another study, the authors found that hamstring length was not related to standing posture. However, those who lengthened their hamstrings after a three-week stretching program did show changes in their lumbar spine and hip movement patterns during forward bending [24]. Therefore, it may be important for farmers to perform hamstring stretches to prevent back pain in in the future.

### Limitations

There are several limitations to this study. First, the present study comprised healthy individuals attending an annual health checkup. The ability to participate in annual health checkups may have led to selection bias with health-conscious and economically affluent patients more likely to be included. Accordingly, the results of the present study may not be applicable to the entire population. However, there is an advantage to the inclusion of asymptomatic participants rather than only patients with low symptomatic back pain as in previous studies. Second, the sample size was small. The number of participants in the present study was limited to 100, which is a small sample size, and some of the comparisons lacked statistical power. Therefore, the study analysis predominantly relied on significant findings rather than comparisons that showed an absence of a statistically significant difference. Third, we did not collect other factors related to agriculture such as type of posture, body part utilized or educational background, and these factors can be unmeasured confounders [25,26]. Jain et al. reported in traditional manual workers, the upper limbs such as shoulders, elbows, hands and fingers are more affected by farming compared to mechanical workers [27]. In addition, full-length images were not present and contributions from the lower extremity joints were not assessed in the current study. Lastly, the degree of degeneration was not assessed, although these findings may be directly related to the outcome.

## 5. Conclusions

We conducted a cross-sectional study comparing spinal radiographic measurements and disability questionnaires between elderly farmers and non-farmers. Farmers were found to have characteristic kyphotic deformities due to anterior translation of the upper spine and loss of LL and thoracic kyphosis rather than as a result of osteoporotic fractures.

## Figures and Tables

**Figure 1 jcm-12-01393-f001:**
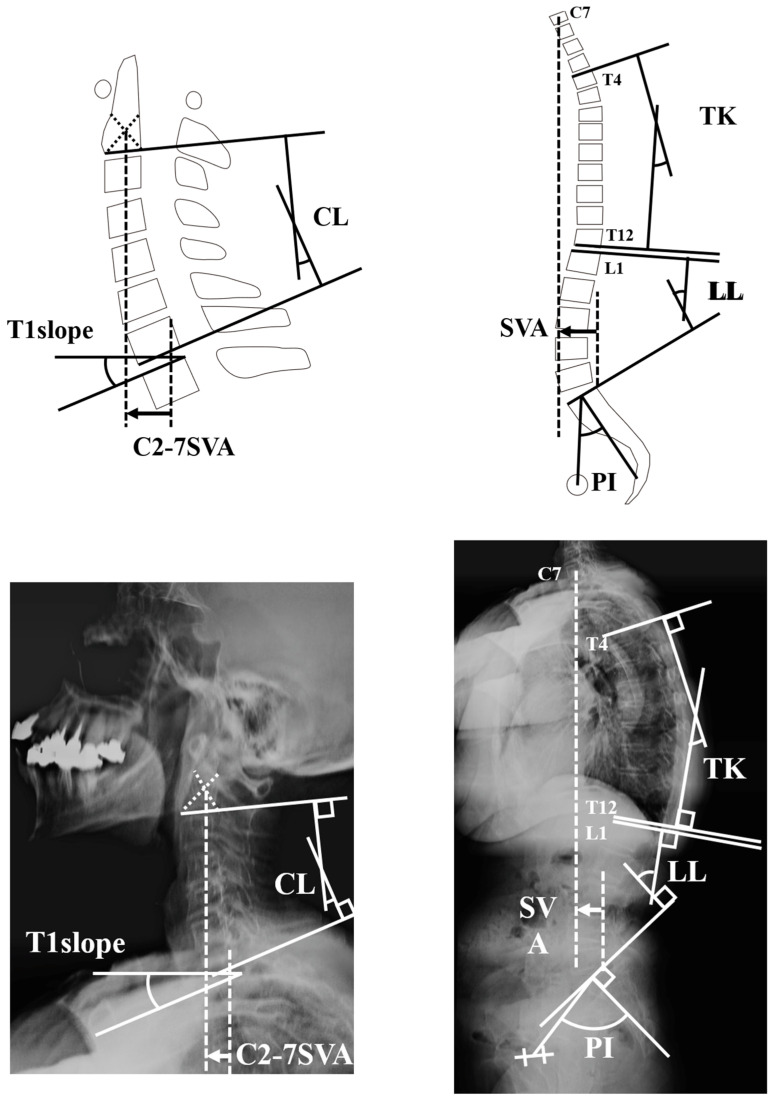
Radiographic parameters.

**Table 1 jcm-12-01393-t001:** Participant demographics.

	*n* (%)	Total (*n* = 100)	Farmer (*n* = 22)	Non-Farmer (*n* = 78)	*p*-Value
Sex	Male	60 (60)	14 (63)	4 (58)	0.69
	Female	40 (40)	8 (36)	32 (41)	
Age, median (IQR) *	71 (65–84)		69 (65–74)	72 (65–84)	0.02
Dominant hand	Right	93 (93)	20 (91)	73 (94)	0.65
	Left	7 (7)	2 (9)	5 (6)	
Family members	Alone	2 (2)	0 (0)	2 (3)	1.00
	With spouse	98 (98)	22 (100)	76 (97)	
Employment status	Employed	52 (52)	22 (100)	30 (38)	<0.01
	Unemployed	48 (48)	0 (0)	48 (62)	

‘Farmer’ includes participants who have worked in agriculture regardless of current involvement in agriculture. Fisher’s exact test * Rank-sum test. Inter Quartile Range (IQR).

**Table 2 jcm-12-01393-t002:** Radiographic parameters and ODI/NDI scores in farmers and non-farmers.

Measurement	Total (*n* = 100)		Farmer (*n* = 22)		Non-Farmer (*n* = 78)		*p*-Value
	Median	Min–Max	Median	Min–Max	Median	Min–Max	
SVA (mm)	12.9	−68–141.8	24.4	−17.9–141.8	9.15	−68–101	0.02
LL (Degree)	43	−25–67	37.5	−25–58	43.5	9–67	0.04
PI (Degree)	47	30–73	47.5	32–66	46	30–73	0.42
PI-LL (Degree)	7	−22–81	14	−11–81	4	−22–36	<0.01
TK (Degree)	36.5	11–62	32.5	11–51	39	17–62	0.02
TK/LL (Degree)	0.9	−1.47–4.33	0.85	−1.47–3	0.91	0.47–4.33	0.45
C2-7SVA (mm)	16.1	−7.4–58.3	16.9	−7.4–35	15.95	−3.5–58.3	0.75
C2-7SVA + SVA (mm)	30.05	−29–169.1	47.65	−17.9–169.1	25.3	−29–159.3	0.03
Cervical LL (Degree)	11.5	−20–43	14	−15–40	9	−20–43	0.59
T1 slope	27	5–46	28	5–38	26.5	8–46	0.88
Diagnosis: Normal Abnormal	61 (61%) 39 (39%)		10 (45%) 12 (55%)		51 (65%) 27 (35%)		0.09
ODI Score	15.6	0–37.8	11.7	0–37.8	6	0–33.3	0.06
NDI Score	12	9–25	13	10–19	12	9–25	0.82

‘Farmer’ includes those who have worked in agriculture regardless of current involvement in agriculture. Rank-sum test. SVA, sagittal vertical axis; LL, lumbar lordosis; PI, pelvic incidence; TK, thoracic kyphosis; CL, cervical lordosis.

**Table 3 jcm-12-01393-t003:** Association among radiographic parameters using Spearman’s correlation among farmers.

	LL	PI	PI-LL	TK	C2-7SVA	C2-7SVA + SVA	Cervical LL	T1 Slope	ODI	NDI
SVA	−0.35	0.16	0.58 *	−0.03	−0.12	0.95 **	0.36	0.29	0.07	0.41
LL		0.36	−0.81 **	0.64 *	0.32	−0.25	−0.06	0.07	−0.37	−0.06
PI			0.16 *	0.17	0.20	0.22	−0.10	0.05	−0.26	0.01
PI-LL				−0.59 *	−0.18	0.51 *	−0.07	−0.09	0.30	0.22
TK					0.19	0.04 *	0.30	0.50 *	−0.38	0.02
C2-7SVA						0.07 *	−0.52 *	0.39	−0.14	0.03
C2-7SVA + SVA							0.22	0.35	0.10	0.33
CL								0.37	−0.17	0.02
T1 slope									−0.51 *	−0.08
ODI										0.16

* *p* < 0.05, ** *p* < 0.001. ODI, Oswestry Disability Index; NDI, Neck Disability Index; SVA, Sagittal vertical axis; LL, lumbar lordosis; PI, pelvic incidence; TK, thoracic kyphosis; CL, cervical lordosis.

**Table 4 jcm-12-01393-t004:** Association among radiographic parameters using Spearman’s correlation among non-farmers.

	LL	PI	PI-LL	TK	C2-7SVA	C2-7SVA + SVA	Cervical LL	T1 Slope	ODI	NDI
SVA	−0.04	0.13	0.22 *	0.28 *	0.07	0.93 **	0.31 *	0.52 **	−0.03	−0.03
LL		0.32 *	−0.64 **	0.51 **	−0.01	−0.02	0.37 **	0.35 *	−0.16	0.09
PI			0.45 **	0.17	−0.16	0.09	0.15	0.22	0.06	0.04
PI-LL				−0.31 *	−0.12	0.17	−0.21	−0.08	0.22	−0.05
TK					0.14	0.35 *	0.48 **	0.71 **	0.11	0.15
C2-7SVA						0.37 **	−0.23 *	0.30 *	0.06	0.11
C2-7SVA + SVA							0.23 *	0.62 **	−0.02	0.00
CL								0.63 **	−0.15	0.04
T1 slope									−0.08	0.00
ODI										0.57 **

* *p* < 0.05 ** *p* < 0.001. ODI, Oswestry Disability Index; NDI, Neck Disability Index; SVA, sagittal vertical axis; LL, lumbar lordosis; PI, pelvic incidence; TK, thoracic kyphosis; CL, cervical lordosis.

**Table 5 jcm-12-01393-t005:** Association between radiographic parameters and ODI/NDI score using Spearman’s correlation among farmers.

	ODI	NDI
SVA	0.07	0.41
LL	−0.37	−0.06
PI	−0.26	0.01
PI-LL	0.30	0.22
TK	−0.38	0.02
C2-7SVA	−0.14	0.03
C2-7SVA + SVA	0.10	0.33
CL	−0.17	0.02
T1 slope	−0.51 *	−0.08
ODI		0.16

* *p* < 0.05. ODI, Oswestry Disability Index; NDI, Neck Disability Index; SVA, sagittal vertical axis; LL, lumbar lordosis; PI, pelvic incidence; TK, thoracic kyphosis; CL, cervical lordosis.

**Table 6 jcm-12-01393-t006:** Association between radiographic parameters and ODI/NDI score using Spearman’s correlation among non-farmers.

	ODI	NDI
SVA	−0.03	−0.03
LL	−0.16	0.09
PI	0.06	0.04
PI-LL	0.22	−0.05
TK	0.11	0.15
C2-7SVA	0.06	0.11
C2-7SVA + SVA	−0.02	0.00
CL	−0.15	0.04
T1 slope	−0.08	0.00
ODI		0.57 **

** *p* < 0.001. ODI, Oswestry Disability Index; NDI, Neck Disability Index; SVA, sagittal vertical axis; LL, lumbar lordosis; PI, pelvic incidence; TK, thoracic kyphosis; CL, cervical lordosis.

## Data Availability

Data are unavailable due to privacy or ethical restrictions.

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
