# Peer review of "A Comparison of Spinopelvic Alignment and Quality of Life between Farmers and Non-Farmers: A Cross-Sectional Population-Based Study in a Japanese Rural Area"

_jcm, 2023, doi:10.3390/jcm12041393_

Round 1
Reviewer 1 Report
Thanks for giving me the opportunity for reviewing this article.
The findings are of some interest but there are some literary and methodological issues as pointed out below:
1. The farm work (especially manual farm work) depends upon several factors such as type of posture (squatting vs. kneeling), body part utilized (single or multiple body part), type of operation (manual or mechanical), education background of farmers, complexity of the operation, heat effect, etc.? I think these factors need to be reflected in literature and discussion section of the paper appropriately. Currently it is not clear which type of farm workers are targeted in the current study.
2. Sampling strategy needs some more description. Also, I feel a pictorial view including the numbers (total, inclusion and exclusion) must give a better overview to the reader.
3. The results are of some interest but the authors could elaborate further on the practical implications of their findings in the discussion section.
4. Some papers related to manual farm working (if the target population belongs to this area) should also be cited and reviewed, such as: https://doi.org/10.1504/IJISE.2019.099776, https://doi.org/10.1080/10773525.2018.1547507, https://doi.org/10.2486/indhealth.2016-0084
5. Please check the literature properly for the count (the sample size for Jain et al. is 138 farmers not 550).
6. Also, there are some run-on, incomplete sentences, which need to be checked before revision submission.
7. The template is not appllied properly.
Author Response
We greatly appreciate the reviewer for the very positive comments. We hope that the revised manuscript will be of interest to the readers of this journal. We would be so happy if the paper is interesting for you and hope that it will be as interesting for the journal audience.
- The farm work (especially manual farm work) depends upon several factors such as type of posture (squatting vs. kneeling), body part utilized (single or multiple body part), type of operation (manual or mechanical), education background of farmers, complexity of the operation, heat effect, etc.? I think these factors need to be reflected in literature and discussion section of the paper appropriately. Currently it is not clear which type of farm workers are targeted in the current study.
Answer: We appreciate the reviewer's suggestion and we agree with that several factors that related to farming may affect the association between being farmer and clinical outcomes. We have described the characteristics of agriculture where the study was conducted as follows.
In the region where our study was conducted, the primary crops produced by farmers were primarily vegetables and rice. Some farmers partially use machinery, but manual labor also made up a large portion of the work process. In vegetable and rice farming, work is mainly done by bending the body against the ground, not by lifting the upper limbs against gravity. While planting and harvesting rice, numerous farmers adopt a posture of maintaining an extended knee position while bending only at the waist, as a result of the burden on the knee joint. These postures are likely to strain the lower back and possibly cause kyphosis deformity of the lumbar spine. (Shin et al. 2004)
Shin, Gwanseob, Yu Shu, Zheng Li, Zongliang Jiang, and Gary Mirka. 2004. “Influence of Knee Angle and Individual Flexibility on the Flexion-Relaxation Response of the Low Back Musculature.” Journal of Electromyography and Kinesiology: Official Journal of the International Society of Electrophysiological Kinesiology 14 (4): 485–94.
- Sampling strategy needs some more description. Also, I feel a pictorial view including the numbers (total, inclusion and exclusion) must give a better overview to the reader.
Answer: Thank you for the clarification about our sampling strategy.
We have added detailed patient selection in our manuscript as follows.
This was a cross-sectional study of 100 local residents aged 65 years or older who underwent a two-day health checkup between October 2020 and January 2021. We included consecutive 100 local residents participating in the health checkup. Ethical approval of the present study was obtained from our institutional review board. All participants provided written consent prior to enrollment. We excluded patients who were unable to undergo standing full spine radiographic examinations due to advanced dementia or lower limb paralysis.
- The results are of some interest but the authors could elaborate further on the practical implications of their findings in the discussion section.
Answer: We appreciate the comment. We further discussed about the insight what we found in the study as follows.
As responded to the comment1, in the cohort that we investigated, manual labor made up a large portion of the work process. The farm work is mainly done by bending the body against the ground, which is likely to strain the lower back and possibly cause kyphosis deformity of the lumbar spine
In order to mitigate the potential for stress on the lumbar region, it is crucial to adopt a posture in which the knee joints are flexed. Adopting such a stance may prevent the exacerbation of lumbar kyphosis resulting from degeneration of the intervertebral discs. (McClure et al. 1997) In another study, the authors found that hamstring length was not related to standing posture. However, those who lengthened their hamstrings after a three-week stretching program did show changes in their lumbar spine and hip movement patterns during forward bending. (Li, McClure, and Pratt 1996) Therefore, it may be important to perform hamstring stretches to prevent back pain in farmers in the future. We have added this to the Discussion. We greatly appreciate this valuable comment.
Li, Y., P. W. McClure, and N. Pratt. 1996. “The Effect of Hamstring Muscle Stretching on Standing Posture and on Lumbar and Hip Motions during Forward Bending.” Physical Therapy 76 (8): 836–45; discussion 845-9.
McClure, P. W., M. Esola, R. Schreier, and S. Siegler. 1997. “Kinematic Analysis of Lumbar and Hip Motion While Rising from a Forward, Flexed Position in Patients with and without a History of Low Back Pain.” Spine 22 (5): 552–58.
- Some papers related to manual farm working (if the target population belongs to this area) should also be cited and reviewed, such as:
Answer: We appreciate the reviewer’s suggestion. We have added the provided studies in our manuscript to explain about the unmeasured confounders as a study limitation.
Lastly, we did not collect other factors related to agriculture such as type of posture, body part utilized, and educational backgrounds, and these factors can be unmeasured confounders. [26,27] Jain et al. reported in traditional manual workers, the upper arm such as shoulder, elbow, hand, and fingers is more affected by farming compared to mechanical workers. [23]
(Limitations)
- Bhardwaj, A.K.; Jain, R.; Dangayach, G.S.; Meena, M.L. Effect of Individual and Work Parameters on Musculoskeletal Health of Manual Agriculture Workers. J. Ind. Syst. Eng. 2019, 32, 56.
- Jain, R.; Meena, M.L.; Dangayach, G.S. Prevalence and Risk Factors of Musculoskeletal Disorders among Farmers Involved in Manual Farm Operations. J. Occup. Environ. Health2018, 1–6
- Jain, R.; Meena, M.L.; Dangayach, G.S.; Bhardwaj, A.K. Risk Factors for Musculoskeletal Disorders in Manual Harvesting Farmers of Rajasthan. Ind. Health 2018, 56, 241–248.
- Please check the literature properly for the count (the sample size for Jain et al. is 138 farmers not 550).
Answer: We apologize for the wrong information we provided and correct them accordingly. We also double checked the previous literatures we have cited in our manuscript.
In 2018, Jain et al. reported a prevalence of lumbar back pain of 71.4% in 138 manual agricultural workers, which was more common than symptoms affecting other parts of the body.[16]
- Also, there are some run-on, incomplete sentences, which need to be checked before revision submission.
Answer: Thank you for the comment. We have asked the native English speaker who has an experience in healthcare field for English proofreading.
- The template is not appllied properly.
Answer: We appreciate the comment, and we have revised our manuscript in align with the template.

Reviewer 2 Report
Better explanation of how the measurements/angles were derived, perhaps with more images as well
I believe that the angles and lines should be better defined, and perhaps with pictures demonstrating how they are measured. Additionally, there are several grammatical errors which should be edited. Other than that, the article is fine. This is why only minor changes are necessary.
Author Response
We greatly appreciate the reviewer for the very positive comments. We hope that the revised manuscript will be of interest to the readers of this journal. The paper was interesting for you and we hope that it will be interesting for the Journal audience.
Better explanation of how the measurements/angles were derived, perhaps with more images as well. I believe that the angles and lines should be better defined, and perhaps with pictures demonstrating how they are measured.
Answer: We appreciate the clarification of our measurement. We have added the measurement we used in Figure using the real x-lay picture in addition to a scheme as follows.
Additionally, there are several grammatical errors which should be edited. Other than that, the article is fine. This is why only minor changes are necessary.
Answer: Thank you for the comment. We have asked the native English speaker who has an experience in healthcare field for English proofreading.

Round 2
Reviewer 1 Report
Thanks for making changes.
Author Response
Thank you for the comment. We hope the current version of manuscript attract reader’s interest in your journal.
